# Decompose to Generalize: Species-Generalized Animal Pose Estimation

**Guangrui Li**[1,2*]**, Yifan Sun**[2]**, Zongxin Yang**[3]**, Yi Yang**[3]
[1]ReLER, AAII, University of Technology Sydney. [2]Baidu Inc.
[3] CCAI, College of Computer Science and Technology, Zhejiang University.
`guangrui.li@outlook.com,sunyf15@tsinghua.org.cn`
`{yangzongxin,yangyics}@zju.edu.cn`

## Abstract

This paper challenges the cross-species generalization problem for animal pose estimation, aiming to learn a pose estimator that can be well generalized to novel species. We find the relation between different joints is important with two-fold impact: 1) on the one hand, some relation is consistent across all the species and may help two joints mutually confirm each other, *e.g.*, the eyes help confirm the nose and vice versa because they are close in all species. 2) on the other hand, some relation is inconsistent for different species due to the species variation and may bring severe distraction rather than benefit. With these two insights, we propose a Decompose-to-Generalize (D-Gen) pose estimation method to break the inconsistent relations while preserving the consistent ones. Specifically, D-Gen first decomposes the body joints into several joint concepts so that each concept contains multiple closely-related joints. Given these joint concepts, D-Gen 1) promotes the interaction between intra-concept joints to enhance their reliable mutual confirmation, and 2) suppresses the interaction between inter-concept joints to prohibit their mutual distraction. Importantly, we explore various decomposition approaches, *i.e.*, heuristic, geometric and attention-based approaches. Experimental results show that all these decomposition manners yield reasonable joint concepts and substantially improve cross-species generalization (and the attention-based approach is the best).

## 1 Introduction

Animal pose estimation (Cao et al., 2019; Li & Lee, 2021b; Mu et al., 2020; Mathis et al., 2021) aims to identify and localize the anatomical joints of animal bodies, and has received increasing attention for its wide application, *i.e.*, biology, zoology, and aquaculture. A critical challenge in realistic animal pose estimation is the cross-species problem, *i.e.*, using the already-learned pose estimator for novel species. Specifically, it is infeasible to collect and annotate all the animal species, because the animal kingdom is a vast group of millions of different species. Under this background, the cross-species generalization is of great value for realistic applications.

This paper tackles the cross-species animal pose estimation from the domain-generalization (DG) viewpoint (*i.e.*, a species is a respective domain) and reveals a unique factor for this cross-species generalization, *i.e.*, the relation between different joints. The joint relation in our view can be visual (*e.g.*, the color relation between neighboring joints), structural (*e.g.*, the nose is under the eyes) and many more. Our focus on the joint relation is different from the popular concern in general domain generalization (Ben-David et al., 2010; Blanchard et al., 2021; 2011; David et al., 2010), which mainly considers the distribution shift between the source and the unseen target domain(s). The generic DG methods usually learn domain-invariant representations (Muandet et al., 2013; Ghifary et al., 2015; Li et al., 2018b;c; Bui et al., 2021; Yang et al., 2021; Gong et al., 2021), enhance the generalizability through meta-learning (Dou et al., 2019; Balaji et al., 2018; Li et al., 2018a; 2019) or data augmentation (Zhou et al., 2021; Shankar et al., 2018; Carlucci et al., 2019; Volpi et al.,

---

*Work done during an internship at Baidu.

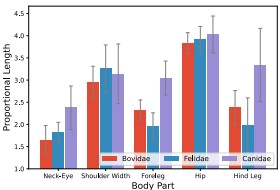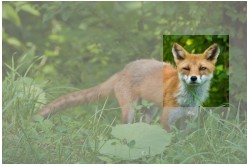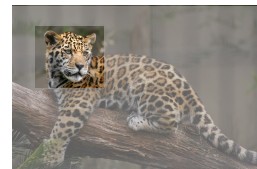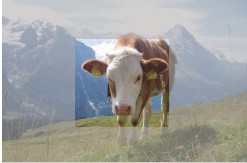

Figure 1: Two reasons that bring domain gap to the joint relation. 1) structural discrepancy: the part lengths (*i.e.* the distances between different joints) may vary for different species (left most); 2) the visual similarities between some different joints are inconsistent for different species, *e.g.*, the visual similarities between faces and other body parts are different for tiger, fox, and the cow.

2018). While these methods are potential for cross-species generalization as well, the joint relation is a unique viewpoint that has never been explored under other DG scenarios.

The importance of joint relation is two-fold: 1) on the one hand, some joint relation is consistent across all the species and is beneficial. With consistent relation, two joints may mutually confirm each other, *e.g.*, the *eye* helps confirm the *nose* and vice versa, because they are consistently close in all species. 2) on the other hand, some joint relation is inconsistent for different species due to species variation and is thus harmful for generalization, *e.g.*, the length of non-rigid body parts such as legs. Such inconsistent relation makes the already-learned mutual confirmation become a severe distraction rather than any benefit. We note that the latter (negative) impact has more or less been recognized by some earlier literature (Cao et al., 2019), while the former (positive) impact was neglected. In contrast to Cao et al. (2019), we argue that both two factors are important and should be considered in combination.

With these two insights, we propose a Decompose-to-Generalize (D-Gen) pose estimation method to break the inconsistent relations while preserving the consistent ones. Specifically, D-Gen first decomposes the body joints into several joint concepts. The decomposition facilitates that each individual concept contains multiple closely-related joints and that the joints in different concepts are far away or prone to inconsistent relations. Given these joint concepts, D-Gen promotes the interaction between intra-concept joints and meanwhile suppresses the interaction between inter-concept joints. The approach for interaction promotion / suppression is very simple: D-Gen splits the top layers of the backbone network into several pose-estimation branches, each one of which is responsible for a corresponding joint concept. Intuitively, the joints in different branches have less interaction, compared to the joints in the same branch. Consequently, D-Gen suppresses the distraction from inconsistent joint relation and yet preserves the beneficial mutual confirmation of consistent joint relation, thus improving cross-species generalization.

We explore three strategies for joint decomposition, *i.e.*, heuristic, geometric and the attention-based manner. The geometric manner clusters the joints based on their geometric distances. The attention-based manner uses the attention mechanism to learn the affinity between joint features and uses the affinity matrix for decomposition. Experimental results show that all these three strategies substantially improve cross-species generalization, validating the effectiveness of our joint decomposition. Another interesting observation is that the attention-based strategy surpasses the other two strategies, indicating that attention-based concepts are better than the concept derived from human intuition (*i.e.*, heuristic) and the pure geometric relation. Since the attention-based approach combines deep feature and geometric priors, its superiority against the geometric manner suggests that there are multiple forms of joint relation beyond the structural relation.

## 2  RELATIVE WORKS

Our work is closely related to two research areas, *i.e.*, pose estimation and domain generalization.

**2D pose estimation for human and animals.** 2D pose estimation refers to identifying all the anatomical joints of bodies for images. There are mainly two paradigms, top-down and bottom-up. The top-down paradigm (Huang et al., 2017; Papandreou et al., 2017; Zhang et al., 2020; Cai et al., 2020; Newell et al., 2016; Moon et al., 2019; Khirodkar et al., 2021) fist detects the person and then localize the joints for each detected person. The bottom-up paradigm (Sun et al., 2019a; Wang et al.,

2019; Geng et al., 2021; Brasó et al., 2021; Jin et al., 2020; Wang et al., 2021; Tang & Wu, 2019; Cao et al., 2017; Kreiss et al., 2019; Newell et al., 2017) first identifies joints in an identity-agnostic manner, then derive poses with different grouping strategies. The top-down solutions are more accurate while requiring higher computation due to the extra detection process, while the bottom-up solutions are more popular for their lower computation cost. Here we adopt the bottom-up solution for its efficiency and simplicity.

In terms of single-species pose estimation, Tang & Wu (2019) performs joint decomposition to strengthen the part with close correlations. However, they only employ the geometric clue inside single species while neglecting the structure variations across species and other forms of joint relations (*e.g.*, visual). Empirical experiments prove it is sub-optimal for the cross-species scenario.

**Domain adaptation and domain generalization.** Domain adaptation aims to transfer the knowledge learned from a labeled source domain to an unlabeled target domain, which has attained remarkable progress in the past decades. Researches in domain adaptation overcome the distribution shift via alignment in the feature space directly (Long et al., 2015; Sun & Saenko, 2016; Tzeng et al., 2014), or employing adversarial training in the input space (Hoffman et al., 2018; Gong et al., 2019; Li et al., 2020) or feature space (Luo et al., 2017; Chen et al., 2019; Ganin & Lempitsky, 2015; Li et al., 2021a; Hu et al., 2022; 2023).

Domain generalization extends to a more universal scenario, where multiple source domains and target domains are available and only source domains are accessible during the training. Researches in this area can be categorized into two streams: one stream seeks to derive domain-invariant representations (Zhao et al., 2020; Muandet et al., 2013; Ghifary et al., 2015; Li et al., 2018b;c; Huang et al., 2020; Seo et al., 2020; Zhang et al., 2021a; Kim et al., 2021), and the other attempts to enhance the generalizability of trained model parameters Wu & Gong (2021); Iwasawa & Matsuo (2021) via meta-learning (Dou et al., 2019; Li et al., 2018a; Zhang et al., 2021b), or data augmentations (Zhou et al., 2021; Carlucci et al., 2019; Li et al., 2021b; Chen et al., 2021; Zhou et al., 2020).

In terms of animal pose estimation, there are already some works considering the domain adaptation problem for it. Mu et al. (2020) and Li & Lee (2021a) focus on the synthetic-to-real domain adaptation inside the same species. Despite making progress, these methods typically mitigate the domain gap induced by the difference in appearance and background, while the structural discrepancies inside the same species are negligible and are not considered as the main obstacle. Cao et al. (2019) made the first attempt to mitigate the structural discrepancies between different species and realize it with an adversarial training paradigm and pseudo labeling. However, we argue it is sub-optimal because their work mainly alleviates the impact resulting from the parts that exhibit large variation across species, while ignoring the parts holding stable inter-joint relationships.

## 3  METHOD

### 3.1  OVERVIEW

**Preliminaries.** From the perspective of domain generalization (DG), each animal species in pose estimation is within an independent domain. Let us assume the training set $\mathcal{D}$ contains $M$ species (domains), *i.e.*, $\mathcal{D} = \{D_1, D_2, ...D_M\}$, forming a joint sample space $\mathcal{X} \times \mathcal{Y}$ ($\mathcal{X}$ is the image space and $\mathcal{Y}$ is the corresponding label space). Specifically, the $i$-th species (domain) $\mathcal{D}_i$ contains $N_i$ samples, *i.e.*, $D_i = \{(x_j^{(i)}, y_j^{(i)})\}_j^{N_i}$. The goal of cross-species pose estimation is to train a pose estimator on $\mathcal{D} = \{D_1, D_2, ...D_M\}$ and then directly apply it to a novel unseen species $D_u (u \notin \{1, 2, ..., M\})$, which requires good cross-domain generalizability. The trained deep model consists of a feature extractor $F$ and an estimator head $G$, which are parameterized by $\phi_F$ and $\phi_G$, respectively.

**The framework** of the proposed D-Gen is illustrated in Fig. 2. The key motivation is that some joint relations are consistent across all the species and are thus beneficial for cross-species generalization, while some other joint relations are inconsistent and harmful. Therefore, D-Gen seeks to break the inconsistent relations while preserving the consistent ones. To this end, D-Gen consists of two stages, *i.e.*, joints decomposition (Section 3.2) and network split (Section 3.3). In the joint decomposition stage, D-Gen divides the body joints into several joint concepts so that each concept contains multiple closely-related joints. Section 3.2 introduces three different decomposition strategies, *i.e.*, heuristic, geometric and attention-based decomposition. Afterward, Section 3.3 pro-

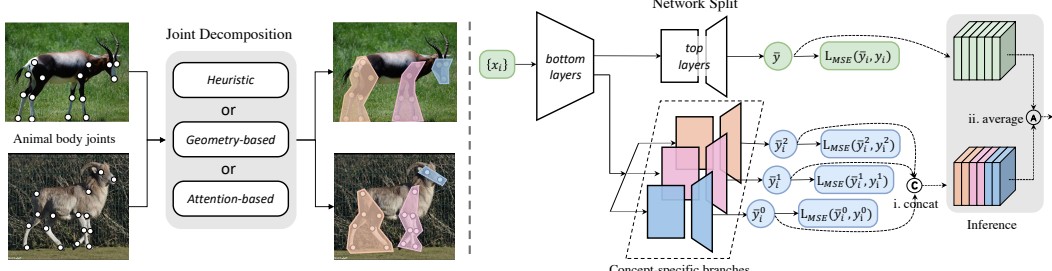

Figure 2: Overview of "Decompose to Generalize" (D-Gen) scheme. D-Gen consists of two stages, *i.e.*, joints decomposition (left) and the sub-sequential network split (right). **1)** In the joints decomposition stage, D-Gen leverages different strategies (*e.g.*, heuristic, geometry-based or attention-based) to divide the body joints into several joint concepts, so that each concept contains closely-related joints (Section 3.2). **2)** Given the decomposed joint concepts, D-Gen correspondingly splits the top layers of the baseline network into multiple concept-specific branches (Section 3.3). This network split suppresses the interaction between inter-concept joints and yet preserves the interaction within each concept. **3)** During inference, D-Gen concatenates the predictions of all the concept-specific branches (step i) and averages them with the baseline prediction (step ii).

poses a very simple network split method that splits the top layers of the backbone network into several concept-specific branches, corresponding to the decomposed joint concepts. Importantly, Sec. 3.3 also investigates the mechanism through gradients analysis and reveals that the network split suppresses the gradient conflict among inconsistently-related joints (*i.e.*, different concepts).

## 3.2 JOINTS DECOMPOSITION

We employ three different concept decomposition strategies, *i.e.*, the heuristic, geometry-based, and the attention-based strategy. The attention-based decomposition is learned based on the baseline network (*i.e.*, we insert an attention module between the feature extractor and the estimation head in the baseline, as in Fig. 3). The heuristic decomposition is drawn from the anatomy knowledge, and the geometry-based decomposition is learned through the geometric distance. Empirically, we find the latter two decomposition strategies are indeed inferior to the attention-based strategy. However, we think the corresponding exploration is valuable for showing that joint decomposition can bring general benefit to cross-species generalization (regardless of the decomposition strategy).

### 3.2.1 ATTENTION-BASED DECOMPOSITION.

We first learn a pixel-to-concept attention Fig. 3 (a) and then use the attention results to infer the decomposition (Fig. 3 (b)).

**Learning pixel-to-concept attention.** As shown in Fig. 3 (a), we append an attention module after the feature extractor to discover the correlation between different joint features. The attention module takes the feature maps $Z \in \mathbb{R}^{hw \times d}$ ($h$ and $w$ are the spatial size of the feature map with $d$ channels) as its input. Meanwhile, we pre-define that there are $k$ concepts ($k$ is a hyper-parameter) and correspondingly provide $k$ concept embeddings, *i.e.*, $E \in \mathbb{R}^{k \times d}$ for learning pixel-to-concept attention. Given the feature maps $Z$ and the concept embeddings $E$, the attention module uses three linear projection to get the `Query`, `Key` and `Value`, respectively, which is formulated as:

$$\texttt{Query} = ZW_q, \texttt{Key} = EW_k, \texttt{Value} = EW_v, \tag{1}$$

where $W_q/W_k/W_v \in \mathbb{R}^{d \times d_l}$ are linear layers that project the inputs into the identical low dimension space. We note that the concept embeddings $E$ and the three linear projections are all learnable through the attention mechanism.

The pixel-to-concept attention is defined as:

$$Z^* = Z + W_m(softmax(\frac{ZW_q(EW_k)}{\sqrt{d_h}})(EW_v)), \tag{2}$$

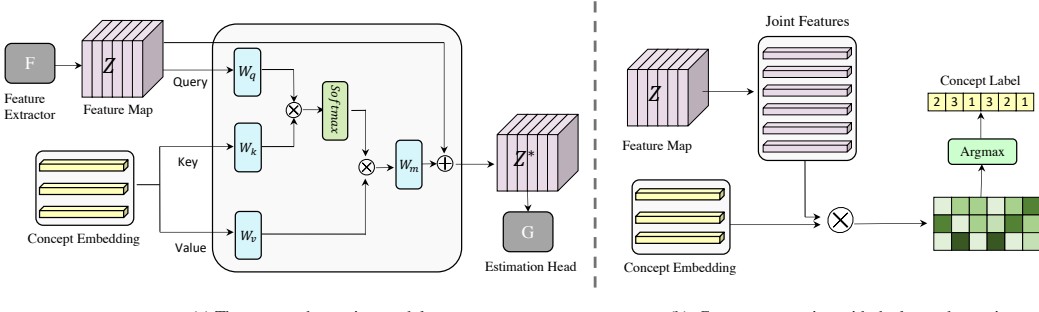

(a) The proposed attention module      (b) Concept generation with the learned attention

Figure 3: **Attention-based decomposition.** Left: Illustration of the proposed attention mechanism. The attention module takes the feature map as query, and a learnable concept embedding as the key for learning the affinity matrix, which is leveraged to promote the inter-joints interactions. Right: Illustration of the concept generation. For the learned feature map, we first extract the joint features based on the predicted joint location. Then with a simple nearest neighbor search with concept embedding, we could obtain the concept label for each joint. The final division of concepts is obtained from the voting from all training images. $\otimes$ denotes the matrix multiplication and $\oplus$ denotes the element-wise sum.

where $W_m$ is a linear layer that maps the dimension of features back to $d$, and the residual connection is added for training stability.

The above attention makes each pixel-level features absorb information from the shared set of concept embeddings. The resulting feature maps $Z^*$ are then fed into the estimation head. We note that while our intention is to use such pixel-to-concept attention as the clues for joint decomposition, the attention actually has another positive side-effect: it already promotes interaction among closely-related joints in an implicit manner. It is because the closely-related joints are likely to absorb information from the same concept embedding(s), thus gaining implicit interactions.

**Decomposition by comparing pixel-wise features to concept embedding.** After the model finishes learning the pixel-to-concept attention, we use the already-learned $k$ concept embeddings $E \in \mathbb{R}^{k \times d}$ to infer the decomposition, as illustrated in Fig. 3 (b). Specifically, given an image and the predicted position of each visible joint, we extract the corresponding joint features from the feature maps $Z$. Let $\hat{z}_{(}i, m)$ denotes the extracted feature of the $m$-th joint from the $i$-th image. According to the cosine similarity between $\hat{z}_{(}i, m)$ and the concept embeddings $E = \{e_j\}_{j=1}^{k}$, the concept-of-interest of the $m$-th joint in the $i$-th image is inferred by:

$$C_{i,m} = \arg \max_{j} \cos < \hat{z}_{i,m}, e_j >, \quad j = 1, 2 ... k, \tag{3}$$

where the $<, >$ denotes the operation of getting the angle of two vectors.

Finally, the concept-of-interest for the $m$-th joint is voted from all the training samples.

### 3.2.2 HEURISTIC AND GEOMETRY-BASED DECOMPOSITION.

In addition to the attention-based decomposition, we explore two alternatives, *i.e.*, the heuristic and geometry-based decomposition.

**Heuristic decomposition.** From the perspective of human knowledge, the joints could be simply divided according to anatomy. Therefore, the heuristic strategy draws the knowledge from the human prior, and decomposes the body joints into three parts, *i.e.*, head, hind leg, and foreleg. Intuitively, the joints within each single part have relatively consistent mutual relations. That being said, experimental results (Sec. 4.2) show that the improvement achieved by heuristic decomposition is relatively small, indicating that relying on intuition is a sub-optimal choice.

**Geometry-based decomposition.** considers the distances between every pair of joints and can be viewed as a pure-structural strategy. The joint distances are derived from the ground-truth joint position and are used to construct an affinity matrix of the joints. Given the affinity matrix, we employ spectral clustering (Ng et al., 2001; Shi & Malik, 2000) to cluster all the joints into multiple concepts. Some examples of the geometry-based concepts are shown in Fig. 6.

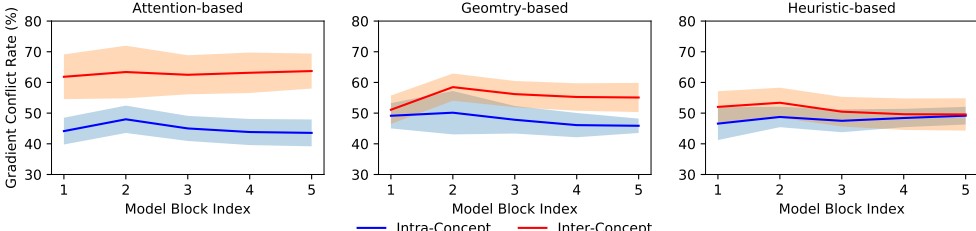

Figure 4: Gradient conflict rate for three decomposition strategies. Generally, the gradient conflict among intra-concept joints is lower than the conflict among inter-concept joints. Moreover, compared with the other two strategies, the attention-based decomposition achieves lower conflict for intra-concept joints, indicating better decomposition. An effective approach to circumvent the gradient conflict is to isolate the optimization of severely-conflicted joints into different branches.

## 3.3 NETWORK SPLIT

Given the joint concepts from a decomposition strategy, D-Gen correspondingly splits the top layers of the baseline network into multiple concept-specific branches, as illustrated in Fig. 2 (the right part). Specifically, the top layers refer to the last $l_{top}$ residual blocks in the backbone (ResNet), while the bottom layers are the rest. Here, we analyze the rationale for this simple network split through gradient analysis.

To be concrete, we use gradient conflict (Yu et al., 2020) to estimate the inter-joint relationships during the optimization. Assuming $\sigma_{ij}$ is the angle between the gradient of joint $i$ and joint $j$, the conflict rate is defined as the proportion of gradients holding different directions, *i.e.*, $cos\sigma_{ij} < 0$. As shown in Fig. 4, the joints from the same concept exhibit higher consistency in the gradient space (*i.e.*, lower conflict rate) while joints from different concepts show a higher possibility to conflict with each other, thus hampering the optimization. Therefore, an intuitive approach to circumvent the gradient conflict is to isolate the optimization of severely-conflict joints (*i.e.*, different concepts) into different network branches, resulting in the proposed network split.

The overall training objective is formulated as:

$$\mathcal{L} = \mathcal{L}_{MSE}(\phi_G(\phi_F(x)), y) + \sum_i^k \mathcal{L}_{MSE}(\hat{\phi}_G^i(\phi_F(x)), y^i). \tag{4}$$

where $\hat{\phi}_G^i$ denotes the feature extractor with the $i$ th concept branch and $y^i$ denotes the ground truth for the $i$ th concept. We use $\mathcal{L}_{MSE}()$ to denote the mean squared error.

During inference, we first concatenate the output of concept-specific branches and then average them with the baseline branch to obtain the final prediction. Notably, before the concatenation, we first sort them according to their original joint index as the joints are not sequentially divided.

**Discussion: The improvement is NOT mainly due to model ensemble.** A side effect of network split is the increase of model size. More specifically, D-Gen splits the top layers of the baseline network and thus can be viewed of the ensemble of multiple concept-specific branches. However, we note that model ensemble (or increasing the model size) is not the major reason for our improvement. An explicit evidence is that: if we replace the joints decomposition strategy with random strategy (*i.e.*, randomly dividing the joints into several concepts, as detailed in Sec. 4.2), the resulting D-Gen only gains very slight improvement, *i.e.*, +1.5% on Animal Pose Dataset (Table 3). In contrast, using attention-based joint decomposition, D-Gen obtains +6.5 % improvement. This observation confirms that model ensemble is only a trivial reason for the improvement of D-Gen.

## 4 EXPERIMENTS

### 4.1 SETUP

**Dataset.** We evaluate our method on two large-scale animal datasets. *AP-10K* (Yu et al., 2021) is a large-scale benchmark for mammal animal pose estimation, and contains 23 animal families and 54

Table 1: Intra-family DG results (AP) on AP-10K (Yu et al., 2021) with the leave-one-out manner. K.C.=King Cheetah, S.L.=Snow Leopard, and A.S. = Argali Sheep. Num.= number of training samples for this species.

| Family | Target | Num. | ERM | Oracle | Oracle-F | Previous SOTA | | | | The proposed D-Gen | | | |
|---|---|---|---|---|---|---|---|---|---|---|---|---|---|
| | | | | | | MixStyle | GradSur | SWAD | Fish | Random | Heuristic | Geometry | Attention |
| Felidae | Bobcat | 151 | 56.4 | 58.4 | 87.1 | 58.1 | 62.0 | 55.1 | 60.4 | 57.9 | 58.8 | 58.2 | **63.2** |
| | Cat | 307 | 30.2 | 54.8 | 79.7 | 40.0 | 28.1 | 27.0 | 34.1 | 35.3 | 37.8 | 35.1 | **41.0** |
| | Cheetah | 148 | 58.6 | 58.7 | 86.3 | 59.0 | 57.9 | 54.1 | 58.5 | 52.2 | 56.0 | 53.1 | **60.1** |
| | Jaguar | 187 | 61.8 | 58.4 | 93.2 | 62.1 | 60.9 | **62.5** | 58.7 | 56.8 | 54.8 | 56.2 | 53.7 |
| | K. C | 22 | 53.1 | 30.6 | 89.8 | 58.6 | 62.0 | 63.6 | 59.0 | 64.6 | 62.9 | **67.9** | 62.1 |
| | Leopard | 142 | 57.8 | 51.1 | 88.1 | 60.4 | 57.7 | **64.3** | 60.4 | 57.7 | 57.1 | 58.2 | 55.4 |
| | Lion | 177 | 40.3 | 45.3 | 84.9 | **45.7** | 34.7 | 40.9 | 41.4 | 40.3 | 36.6 | 37.4 | 38.8 |
| | Panther. | 106 | 41.5 | 55.4 | 86.2 | 45.5 | 40.7 | 38.6 | 37.2 | 46.2 | 43.9 | 44.3 | **48.4** |
| | S.L. | 73 | 65.4 | 54.8 | 95.4 | 69.0 | **73.7** | 67.3 | 61.0 | 61.2 | 62.0 | 63.9 | 64.9 |
| | Tiger | 144 | 58.9 | 68.9 | 87.6 | 57.1 | 59.3 | 47.4 | 57.8 | 56.2 | **64.2** | 62.3 | 62.4 |
| | Average | − | 52.4 ± 0.6 | 53.6 ± 0.3 | 87.8 | 53.1 ± 0.9 | 53.7 ± 0.5 | 52.1 ± 0.6 | 53.9 ± 1.2 | 52.9 ± 1.0 | 53.4 ± 0.9 | 53.7 ± 0.3 | **55.0 ± 0.5** |
| Ursidae | Black Bear | 39 | 30.0 | 43.2 | 85.2 | 37.7 | 37.5 | 30.6 | 24.8 | 26.8 | 36.1 | **41.1** | 40.7 |
| | Brown Bear | 171 | 15.5 | 42.2 | 83.1 | 14.9 | 17.6 | 24.1 | 23.7 | 15.8 | **28.0** | 20.4 | 24.8 |
| | Panda | 164 | 9.0 | 46.9 | 79.0 | 13.9 | 7.5 | 11.1 | 17.5 | 13.8 | 13.2 | 12.7 | **14.5** |
| | Polar Bear | 156 | 10.6 | 47.1 | 85.1 | 9.1 | 11.5 | 13.3 | 17.6 | 9.1 | **21.2** | 8.3 | 18.7 |
| | Average | − | 18.8 ± 0.4 | 44.9 ± 0.2 | 83.1 | 18.9 ± 0.3 | 18.6 ± 0.5 | 19.8 ± 0.4 | 20.4 ± 0.5 | 18.8 ± 0.9 | 21.0 ± 0.6 | 20.6 ± 0.4 | **24.7 ± 0.4** |
| Bovidae | Antelope | 298 | 51.3 | 63.1 | 88.4 | 52.6 | 53.4 | 52.4 | 56.2 | 56.0 | 53.4 | 53.9 | **60.9** |
| | A.S | 268 | 69.7 | 70.1 | 95.6 | 71.3 | 74.1 | 67.6 | 75.1 | 72.6 | 73.6 | 74.3 | **77.3** |
| | Bilson | 208 | 42.8 | 51.6 | 91.8 | 49.8 | 43.6 | 46.2 | 47.7 | 46.1 | 48.8 | 45.8 | **49.8** |
| | Buffalo | 228 | 61.0 | 71.6 | 91.4 | 57.5 | 64.8 | 63.7 | 62.3 | 60.0 | 61.1 | 62.9 | **68.1** |
| | Cow | 228 | 46.6 | 43.9 | 82.1 | 50.7 | 46.1 | 49.7 | 46.9 | 48.9 | 49.7 | 48.8 | **51.8** |
| | Sheep | 355 | 41.7 | 57.4 | 85.0 | 41.0 | 42.1 | **45.1** | 41.6 | 40.6 | 41.0 | 41.2 | 40.1 |
| | Average | − | 52.2 ± 0.6 | 59.6 ± 0.3 | 89.1 | 53.8 ± 0.6 | 54.1 ± 0.4 | 53.9 ± 0.5 | 55.0 ± 0.4 | 53.8 ± 0.8 | 54.6 ± 0.3 | 54.5 ± 0.4 | **57.9 ± 0.4** |

species following the taxonomic rank. It annotates 10,015 images and provides annotations with **17 keypoints**. *Animal Pose Dataset* (Cao et al., 2019) collects and annotates 5 species, *i.e.*, cat, dog, sheep, cow, and horse, from 3000+ images, where both bounding boxes and keypoints annotated (**20 keypoints**) are provided. *Animal Kingdom* (Ng et al., 2022) is another dataset with a higher diversity that includes mammals, fishes, birds, amphibians, and reptiles with **23 keypoint** annotations.

**Evaluation.** Follow Sun et al. (2019b), the evaluation metric is based on Object Keypoint Similarity (OKS). We report the mean average precision (AP) at OKS=0.50, 0.55,...0.90, 0.95. The mean value and standard error over three *random* runs are reported in all main results.

**Implementation details.** We start from ResNet50 (He et al., 2016) with the backbone pretrained on ImageNet (Deng et al., 2009). The batch size is set to 64, and the learning rate for the first stage and the second is set to $5 \times 10^{-4}$ and $5 \times 10^{-5}$, respectively. We optimize the model with Adam for 210 epochs, where the learned rate decrease ($\times 10^{-1}$) at 170 and 200, respectively. The size of the input image is 256 × 256 and the heatmap is with size 64 × 64. The number of the concept-specific blocks is set to 2, and $k$ is set to 3 for all transfer tasks.

## 4.2 MAIN RESULTS.

We evaluate our approach under two domain generalization (DG) scenarios, *i.e.*, intra-family DG and inter-family DG, where the latter involves larger domain gaps regarding both the visual and structural knowledge. Following the common practice, we perform domain generalization with the leave-one-out setting, *i.e.*, selecting one species / family as the target domain and the rest as the source domains. Since these two datasets are relatively new and only a few results have been reported, we re-implement representative DG methods, *i.e.*, Fish (Shi et al., 2022), SWAD (Cha et al., 2021), MixStyle (Zhou et al., 2021), and GradSur (Mansilla et al., 2021) for comparison and we adopt ERM (empirical risk minimization that trains on all source domains), Oracle (training on target domains), and Oracle-Full (train on both source and target domains) as baselines. Moreover, we use a random decomposition in addition to the three discussed decomposition strategies to show that model ensemble (through random-decomposed branches) does not promise improvement.

**Intra-family DG.** For intra-family domain generalization, we evaluate our approach on the three largest families in AP-10K, *i.e.*, Bovidae, Canidae, and Ursidae. The results are summarized in Table 1, from which we draw three observations.

**1)** Comparing "Previous SOTA" against the "ERM" baseline, we find the improvement achieved by previous state-of-the-art DG methods is very trivial (if there is any). We infer it is because these methods mainly focus on the style gap of the source and target domain, and are not capable to tackle the distribution shift in terms of structural and visual relations.

**2)** Comparing the proposed D-Gen against the "ERM" baseline, we observe that all the three decomposition strategies (heuristic, geometry-based, and attention-based) achieve consistent improvement.

Table 2: Inter-family DG results (AP) on AP-10K (Yu et al., 2021) with the leave-one-out manner. Cerc.=Cercopithecidae.

| Target | Num. | ERM | Oracle | Oracle-F | Previous SOTA | | | | The proposed D-Gen | | | |
|---|---|---|---|---|---|---|---|---|---|---|---|---|
| | | | | | MixStyle | GradSur | SWAD | Fish | Random | Heuristic | Geometry | Attention |
| Bovidae | 1467 | 48.2 | 64.8 | 90.3 | 49.6 | 47.0 | 48.8 | 49.3 | 48.6 | 46.0 | 49.9 | **52.1** |
| Felidae | 1457 | 39.7 | 64.4 | 87.3 | 43.1 | 34.2 | 42.4 | 39.8 | 43.0 | 42.8 | 41.9 | **47.7** |
| Canidae | 1130 | 50.2 | 65.9 | 88.8 | 48.2 | 46.8 | 53.6 | 50.6 | 53.3 | 49.7 | 53.9 | **54.8** |
| Ursidae | 530 | 30.3 | 68.8 | 83.7 | 31.3 | **34.5** | 31.4 | 32.3 | 31.5 | 30.7 | 28.4 | 31.1 |
| Cerc. | 623 | 19.9 | 67.9 | 84.2 | 20.3 | 19.1 | 21.8 | 24.4 | 24.7 | **40.8** | 23.0 | 27.2 |
| Equidae | 482 | 37.8 | 64.2 | 82.6 | 37.9 | 34.2 | 40.5 | 38.8 | 39.7 | 40.6 | **42.8** | 39.0 |
| Hominidae | 345 | 39.6 | 66.9 | 83.1 | 38.4 | 32.4 | 38.7 | 39.7 | 38.4 | 40.0 | **46.1** | 45.6 |
| Average | − | 37.7 ± 0.4 | 66.1 ± 0.2 | 85.7 | 38.4 ± 0.4 | 35.5 ± 0.5 | 39.8 ±0.6 | 39.3 ± 0.5 | 39.6 ± 0.9 | 40.0 ± 0.7 | 40.9 ± 0.7 | **42.9 ± 0.6** |

Table 3: Results (AP) on Animal Pose Dataset (Cao et al., 2019) with the leave-one-out manner.

| Target | Num. | ERM | Oracle | Previous SOTA | | | | The proposed D-Gen | | | |
|---|---|---|---|---|---|---|---|---|---|---|---|
| | | | | MixStyle | GradSur | SWAD | Fish | Random | Heuristic | Geometry | Attention |
| Cow | 1214 | 22.6 | 36.8 | 25.0 | 24.4 | 18.8 | 27.8 | 26.7 | **29.9** | 25.7 | 29.6 |
| Sheep | 980 | 21.6 | 33.0 | 23.2 | 22.0 | 21.6 | 21.2 | 20.4 | 26.1 | 25.4 | **28.4** |
| Horse | 651 | 22.7 | 37.5 | 27.8 | 26.2 | 26.1 | 25.4 | 28.5 | 31.7 | **33.6** | 31.8 |
| Cat | 614 | 26.3 | 30.7 | 27.8 | 24.8 | 25.1 | 26.3 | 24.3 | 23.9 | 23.2 | **28.1** |
| Dog | 502 | 21.2 | 31.1 | 25.1 | 28.2 | 22.2 | 26.2 | **24.4** | 23.7 | 25.0 | 26.4 |
| Average | − | 23.4 ± 0.4 | 33.8 ± 0.3 | 25.8 ± 0.6 | 23.9 ± 0.4 | 22.7 ± 0.5 | 25.4 ± 0.4 | 24.9 ± 0.8 | 27.1 ± 0.4 | 26.6 ± 0.5 | **28.9 ± 0.4** |

For example, within the Bovidae family, these three methods surpass the ERM baseline by +2.3%, +3.2%, and +5.2%, respectively. This observation validates the effectiveness of D-Gen.

**3)** Comparing the three decomposition strategies against each other and the additional "Random" strategy, we observe that the attention-based strategy is the best among all. It indicates that the deep network is capable to discover better joint concepts than human intuition and pure-geometric knowledge. Another important observation is that the "Random" strategy barely achieves any improvement over the baseline. It indicates that D-Gen barely benefits from the model ensemble, because the random strategy already takes the advantage of model ensemble.

**4)** Under circumstances with fewer labeled samples, domain generalization can serve as a powerful baseline for pose estimation. For example, our method surpasses the Oracle with 4.8 % on the species Bobcat from the family Felidae, where only 150 images are provided with annotations.

**Inter-family DG.** We further evaluate the proposed D-Gen under the inter-family DG scenario. Without loss of generality, we select several animal families with relatively large diversity for each dataset (Table 2 for AP-10K and Table 3 for Animal Pose Dataset). We observe our method maintains its superiority against previous methods, which is consistent with the observation under the intra-family scenario. Moreover, we observe that under the inter-family scenario, the achieved results are lower than the intra-family scenario. It confirms our intuition that inter-family generalization is more challenging due to larger cross-species distribution shifts.

Table 4: Results (PCK@0.05) on Animal Kingdom Dataset (Ng et al., 2022) with the leave-one-out manner. AM=Amphibians.

| Family | Mammals | AM | Reptiles | Birds | Fishes | Avg. |
|---|---|---|---|---|---|---|
| ERM | 11.5 | 15.0 | 11.4 | 13.2 | 12.4 | 12.7 |
| Oracle | 37.3 | 69.1 | 66.5 | 52.0 | 45.5 | 54.1 |
| SWAD | 13.4 | 18.3 | 15.5 | 17.4 | 12.6 | 15.4 |
| Fish | 16.8 | 17.9 | 17.7 | 18.1 | 16.6 | 17.4 |
| Heur. | 16.6 | 16.9 | 18.1 | 18.8 | 15.1 | 17.1 |
| Geom. | 17.1 | 18.5 | 18.9 | 19.1 | 16.1 | 17.9 |
| Att. | 18.3 | 19.0 | 19.4 | 19.9 | 18.8 | **19.1** |

## 4.3 ABLATION STUDIES

In this section, we conduct ablation studies under two scenarios, *i.e.*, intra-family DG (within the Bovidae family, in particular) and the inter-family DG, to further investigate the proposed method.

**Different number of concepts ($k$).** In Fig. 5 (a)(b), we report the results with varying $k$ to evaluate its influence. With $k$ increasing, these decomposition solutions show different tendencies, *i.e.*, the attention-based variant only fluctuates in a small range, while the other two show degraded performance. Such a phenomenon verifies that the attention-based solution is not sensitive to the selection of $k$. Further investigation reveals that, as the $k$ increased, the concept embedding still derives concept groups with a stable number, *i.e.*, from 3 to 5. The reason is that, when $k$ is large, only part of the concept embeddings attend and modulate the inter-joint interactions while others not.

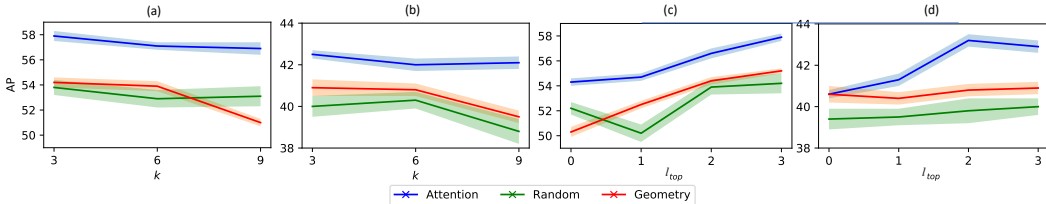

Figure 5: The impact of $k$ (the number of concepts) and $l_{top}$ (the number of concept-specific blocks) with error bars. Both the intra-family (a and c) and the inter-family scenarios (b and d) are presented.

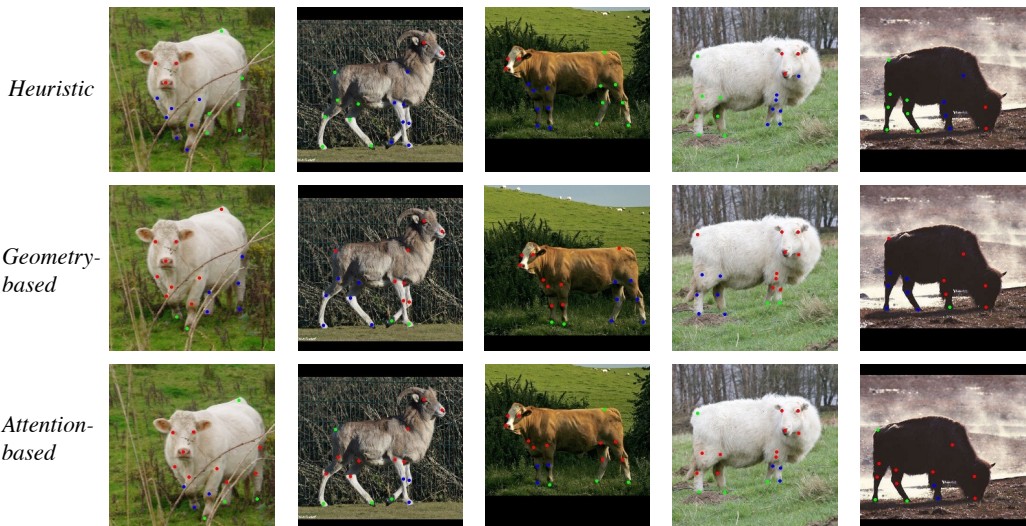

Figure 6: Concept visualization on AP-10K, where different colors denote different mined concepts.

**How does $l_{top}$ (number of concept-specific blocks) influence the generalization results?** In Fig. 5 (c)(d), we present the sensitivity analysis to the number of concept-specific blocks ($l_{top}$). As we could observe, with increasing $l_{top}$, the performance improves constantly, justifying the necessity of the network split pipeline. Especially, even with the shared backbone, our method still achieves very competitive results, *i.e.*, 40.6 for inter-family DG, proving its superiority against previous methods. In contrast, the other two solutions only achieve very limited improvement with the increasing $l_{top}$. The reason is that, as illustrated in Fig. 4, the joints of their mined concepts still exhibit high conflict with each other, which is the network split cannot mitigate.

**Visualization of the learned concepts.** Fig. 6 compares three joint decomposition strategies through visualization. For a clear comparison, we adopt three concepts (painted in red, blue, and green color) for all the strategies. We observe that the attention-based decomposition tends to group joints from comparably distant positions into a concept, benefiting from the learning of long-range dependencies with the attention module. In contrast, the heuristic and geometry-based solutions mainly focus on the local structures and tend to associate joints with their neighbors.

## 5 CONCLUSION

In this paper, we propose a "decompose to generalize" (D-Gen) scheme for cross-species animal pose estimation. In contrast to generic domain generalization methods, D-Gen focuses on the joint relation for improving the cross-species generalization. Specifically, D-Gen decomposes the body joints into several joint concepts, so that the joints in a single concept have relatively consistent relations. Based on the joint concepts, D-Gen splits the feature extractor into multiple concept-specific branches. This simple network split suppresses the inconsistent interactions between inter-concept joints and yet maintains the consistent interactions between intra-concept joints. Experimental results validate the effectiveness of D-Gen. We hope our work on cross-species pose estimation can provides a new viewpoint for understanding the domain generalization problem.

ETHICS STATEMENT

**Our method does not induce bias but cannot alleviate the bias from the training data.** Thus, with the proposed method, the model trained on source domains may still preserve the bias from the source data, hence resulting inaccurate estimation results in the target domain.

**Our method may fail under extreme scenarios with severe occlusion.** The proposed method mainly assumes an ideal scenario with no or slight occlusion. However, wild environments and animal habitats typically have various terrains and vegetation, which possibly causes severe occlusion. Our method may fail to deliver reliable pose estimations under such extreme circumstances.

**Our method assumes a well-align joint space.** Under the diverse distribution of animal species, our method may fail to generalize to species with novel keypoints. Due to the limited diversity of current animal datasets, we cannot provide further analysis on the generalization to novel keypoints or species, *e.g.*, from four-leg animals to birds, but hope to explore it in future works.

**We hope our work can help protect the animals on our planet.** Animal pose estimation can serve as an important tool for analyzing animal behaviors and protecting them. However, training a reliable pose estimator requires massive fine-annotated datasets, which is unrealistic for wild animals, especially for rare species. Our method can help overcome this obstacle by learning species-generalizable representations, thus reducing the need for labor-extensive collections and annotations.

REPRODUCIBILITY STATEMENT

The proposed method is reproducible. We have provided the PyTorch-style code for the proposed attention module, and the detailed training strategies in the main text and the appendix.

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

Table 5: Ablation studies on attention module for intra-family domain generalization (family Bovidae). Experiments here are conducted on AP-10K (Yu et al., 2021) with the leave-one-out manner. A.S. = Argali Sheep.

| Decomposition | With attention | Target Domain | | | | | | Average |
|---|---|---|---|---|---|---|---|---|
| | | Antelope | A.S. | Bilson | Buffalo | Cow | Sheep | |
| Random | | 56.0 | 72.6 | 46.1 | 60.0 | 48.9 | 40.6 | 53.8 |
| | ✓ | 52.6 (↓) | 67.7 (↓) | 51.1 (↑) | 60.0 (−) | 43.7 (↓) | 42.8 (↑) | 53.0 (↓) |
| Heuristic | | 53.4 | 73.6 | 48.8 | 61.1 | 49.7 | 41.0 | 54.6 |
| | ✓ | 49.6 (↓) | 69.8 (↓) | 48.4 (↓) | 58.9 (↓) | 47.8 (↓) | 39.7 (↓) | 52.4(↓) |
| Geometry | | 53.9 | 74.3 | 45.8 | 62.9 | 48.8 | 41.2 | 54.5 |
| | ✓ | 55.4 (↑) | 70.3 (↓) | 47.0 (↑) | 63.8 (↑) | 48.8 (−) | 44.8 (↑) | 55.1 (↑) |
| Attention | ✓ | 60.9 | 77.3 | 49.8 | 68.1 | 51.8 | 40.1 | 57.9 |

Table 6: Ablation studies on attention module for inter-family domain generalization. Experiments here are conducted on AP-10K (Yu et al., 2021) with the leave-one-out manner. Cerc.=Cercopithecidae.

| Decomposition | With attention | Target Domain | | | | | | | Average |
|---|---|---|---|---|---|---|---|---|---|
| | | Bovidae | Felidae | Canidae | Ursidae | Cerc. | Equidae | Hominidae | |
| Random | | 48.6 | 43.0 | 53.3 | 31.5 | 24.7 | 39.7 | 38.4 | 39.6 |
| | ✓ | 49.3 (↑) | 42.0 (↓) | 51.2 (↓) | 32.0 (↑) | 25.6 (↑) | 40.6 (↑) | 37.7 (↓) | 39.8 (↓) |
| Heuristic | | 46.0 | 42.8 | 49.7 | 30.7 | 40.8 | 40.6 | 40.0 | 40.0 |
| | ✓ | 46.6 (↑) | 43.9 (↑) | 49.2 (↓) | 32.8 (↑) | 29.7 (↓) | 42.2 (↑) | 41.3 (↑) | 40.8 (↑) |
| Geometry | | 49.9 | 41.9 | 53.9 | 28.4 | 23.0 | 42.8 | 46.1 | 40.9 |
| | ✓ | 50.8 (↑) | 42.6 (↑) | 54.8 (↑) | 36.4 (↑) | 25.2 (↑) | 39.9 (↓) | 42.6 (↓) | 41.8 (↑) |
| Attenion | ✓ | 52.1 | 47.7 | 54.8 | 31.1 | 27.2 | 39.0 | 45.6 | 42.9 |

# APPENDIX

## A  ADDITIONAL ANALYSIS AND RESULTS

**Ablation of the pixel-to-concept attention mechanism.** We note that the pixel-to-concept attention mechanism actually has two impacts on the attention-based decomposition, *i.e.*, 1) it learns clues for joint decomposition, and 2) it incurs interaction between joints and concept proxies, while the other two decomposition strategies do NOT have such effect. Therefore, although Table 1 and Table 2 in the manuscript show that the attention-based decomposition achieves the largest improvement among all the four strategies, we are not clear what is the exact reason for its superiority. We answer this question by adding the same attention module into the network under the other three decomposition strategies (without changing their joint decomposition results). The results are summarized in Table 5 (intra-family) and Table 6 (inter-family), from which we draw two observations as below:

First, we observe that adding the attention module for joint interaction does not promise improvement. For example, it slightly compromises the random and the heuristic decomposition ($53.8\rightarrow53.0$ and $54.6\rightarrow52.4$) under the intra-family DG scenario. Second, for geometry-based decomposition, adding the attention module leads to a slight improvement, *i.e.*, + 0.6 for geometry-based solution in Table 5. However, it is worth noting that, even with the attention, it still cannot compete with the attention-based decomposition. Combining these two observations, we conclude that the superiority of the attention-based solution should not be mainly ascribed to the pixel-to-concept interactions but to better joint concepts.

**What if treat all joints as one concept or treat each joint as individual concepts?** In Table 7, we complement the results when manually set $k = 1$ and $k = 17$, which treats all joints one concept or treat each joint as one concept. Despite the network split, the manual concept division barely improves the performance over the ERM baseline, which again justifies the effectiveness of the proposed D-Gen paradigm. These results also validate that simply increasing the model capacity cannot promise improvement in the generalization.

Table 7: DG results (AP) on AP-10K (Yu et al., 2021) under varying $k$ (number of concepts) with D-Gen paradigm or manual setting.

| | | D-Gen | | | Manual | |
|---|---|---|---|---|---|---|
| | Decom. | 3 | 6 | 9 | 1 | 17 |
| | Attention-based | 57.9 | 57.1 | 56.9 | | |
| Intra-DG | Geometry-based | 54.2 | 53.9 | 51.0 | 52.8 | 51.9 |
| | Random | 53.8 | 52.9 | 53.1 | | |
| | Attention-based | 42.5 | 42.0 | 42.1 | | |
| Inter-DG | Geometry-based | 40.9 | 40.8 | 39.5 | 38.0 | 38.2 |
| | Random | 39.6 | 40.3 | 38.8 | | |

Table 8: Comparison on the training time of each iteration.

| Method | ERM | MisStyle | GradSur | SWAD | Fish | Ours |
|---|---|---|---|---|---|---|
| Time of iteration (ms) | 250 | 251 | 310 | 290 | 2246 | 280 |

**Comparison on more datasets.** In Table 10, we present the generalization results to Horse-C dataset (Mathis et al., 2021). Following their evaluation protocol, we validate the effectiveness of the proposed method on this benchmark and report the results, and again attains superior results.

**Analysis on joint feature extraction** In Table 11, we testify the sensitivity to the feature extraction process, i.e., localizing the joint features with ground truth or the prediction. The negligible difference verifies that our method is robust to the choice on clues for feature extraction.

**Analysis on model capacity** In Table 9, we present the analysis of the model capacity, *i.e.*, FLOPs and inference speed. And in Table 8, we compare the time for each training iteration . Apparently, our method does not require longer time even with two stages when compared with some previous SOTAs. This is because some of them employ extra inner steps during the optimization. For example, Fish (Shi et al., 2022) requires to perform $n$ inner steps to aggregate the gradients of $n$ domains independently.

**Evaluation of uncertainty.** In Table 12, we repeat our experiments on two more data splits with three different random seeds. Under such variation, we re-evaluate the uncertainty in two scenarios, i.e., intra-family DG within Bovidae and inter-family DG on AP-10K. The results show consistent uncertainty with the main paper on both scenarios.

Table 9: Analysis on the model capacity.

| Model configuration | FLOPs (G) | Inference speed(MS) |
|---|---|---|
| Original Network (ResNet-50) | 21.0 | 9.2 |
| with network split | 24.8 | 12.4 |

Table 10: PCK@0.3(%) for out-of-distribution generalization to Horse-C (Mathis et al., 2021) (FF=front foot; HF = Hind foot; HH = Hind Hock).

| Method | Nose | Eye | Shoulder | Wither | Elbow | NearFF | OffFF | Hip | NearHH | NearHF | OffHF | Average |
|---|---|---|---|---|---|---|---|---|---|---|---|---|
| Mathis et al. (2021) | 68.2 | 73.6 | 85.4 | 85.8 | 88.1 | 72.6 | 70.2 | 89.2 | 85.7 | 77.0 | 74.1 | 79.1 |
| D-Gen (attention) | 67.9 | 76.4 | 86.1 | 83.8 | 88.3 | 79.6 | 74.3 | 90.1 | 86.6 | 79.3 | 76.6 | 80.9 |

Table 11: Results with different clues for joint feature extraction. Experiments here are under the inter-family DG scenario on AP-10K (Yu et al., 2021). Cerc.=Cercopithecidae.

| Extraction clue | Target Domain | | | | | | | Average |
|---|---|---|---|---|---|---|---|---|
| | Bovidae | Felidae | Canidae | Ursidae | Cerc. | Equidae | Hominidae | |
| Ground truth | 53.1 | 47.2 | 52.3 | 35.9 | 28.7 | 47.5 | 46.6 | 43.2 |
| Prediction | 52.1 | 47.7 | 54.8 | 31.1 | 27.2 | 39.0 | 45.6 | 42.9 |

Table 12: More random runs for uncertainty evaluation. Experiments here are conducted on AP-10K (Yu et al., 2021) in the leave-one-out manner. The intra-family DG is performed on the family Bovidae.

| Transfer | Data Split 1 | | | | Data Split 2 | | | |
|---|---|---|---|---|---|---|---|---|
| | Random run 1 | Random run 2 | Random run 3 | Average | Random run 1 | Random run 2 | Random run 3 | Average |
| Intra-Family DG | 57.91 | 57.13 | 58.03 | $57.69 \pm 0.49$ | 57.47 | 58.34 | 57.32 | $57.81 \pm 0.47$ |
| Inter-Family DG | 43.13 | 42.14 | 42.63 | $42.63 \pm 0.50$ | 42.92 | 43.31 | 41.98 | $42.73 \pm 0.68$ |

**More qualitative comparisons between different joint decomposition strategies.** In Fig. 7, we provide more visualizations of the mined concepts with different decomposition strategies, and we make the following observations:

1) Despite being intuitively reasonable, heuristic decomposition may induce ambiguity between some hard-to-distinguish joints. For example, the left and right forelegs are assigned to the same concept, while distinguishing them within the same concept might not be easy.

2) For the geometry-based solution, we notice that it tends to focus on and identify closely-related structures in the geometric space, *i.e.*, the forelegs and the hind legs.

3) In Fig. 7 (c), we find that the left eye and the right eye are assigned to different concepts. This result may seem counter-intuitive at first glance but is actually reasonable from a closer look. It is because the left and right eyes are very small and hard to distinguish. Therefore, associating the left and right eyes with some different easy-to-distinguish joints helps to discriminate them, as well. On the contrary, both the heuristic and geometry-based strategies assign the left and right eyes to the same concept and are actually inferior.

**Qualitative comparisons on pixel-to-concept attention.** To better understand the pixel-to-concept attention, we present the visualization of the learned attention maps in Fig. 8. As we could observer, each concept embedding apparently favors specific regions, and such preference is consistent over different species. Such a phenomenon verifies that the proposed attention mechanism indeed encourages the inter-joint relationships that can benefit the generalization.

**Qualitative comparison with previous solutions on pose estimation results.** In Fig. 9 and Fig. 10, we present the qualitative results on intra-family and inter-family DG, respectively, and compare with previous solutions. Compared with the previous solution, the proposed D-Gen paradigm can deliver more accurate estimations, proving the effectiveness of the proposed method. Concretely, the larger variation in appearance and shape renders previous solutions less effective, especially on the joints from non-rigid parts, *e.g.*, legs. In contrast, our method, especially the attention-based variant, shows stronger robustness to them and delivers more accurate estimations on both intra-DG and inter-DG scenarios.

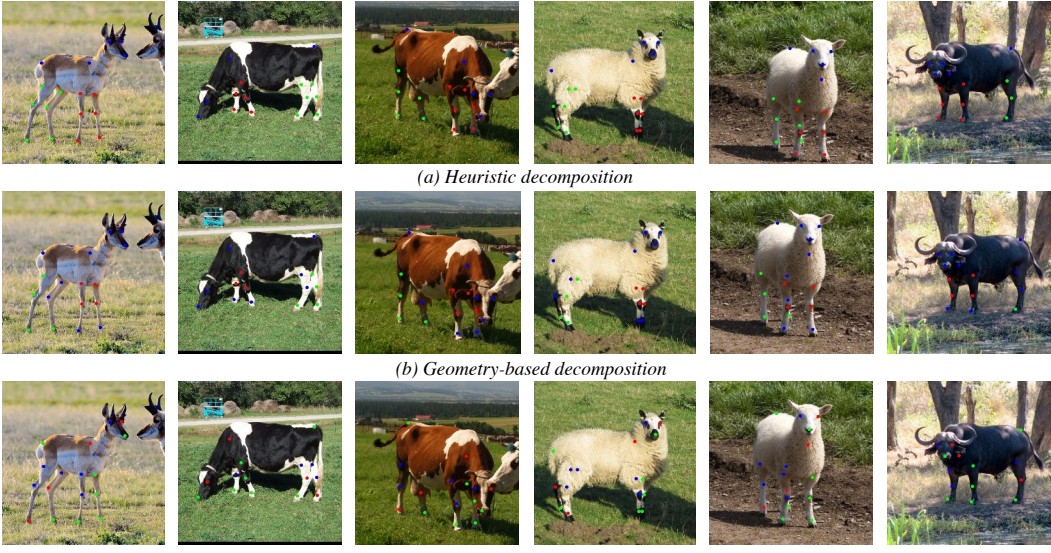

Figure 7: Concept visualization on AP-10K (Yu et al., 2021), where different mined concepts are in different colors. A noticeable observation is that the attention-based strategy assigns the left and right eyes to different concepts. This decomposition result seems counter-intuitive but is actually reasonable: it associates the hard-to-distinguish left and right eyes with different easy-to-distinguish joints, so that the latter joints provide clues for distinguishing the left and right eyes.

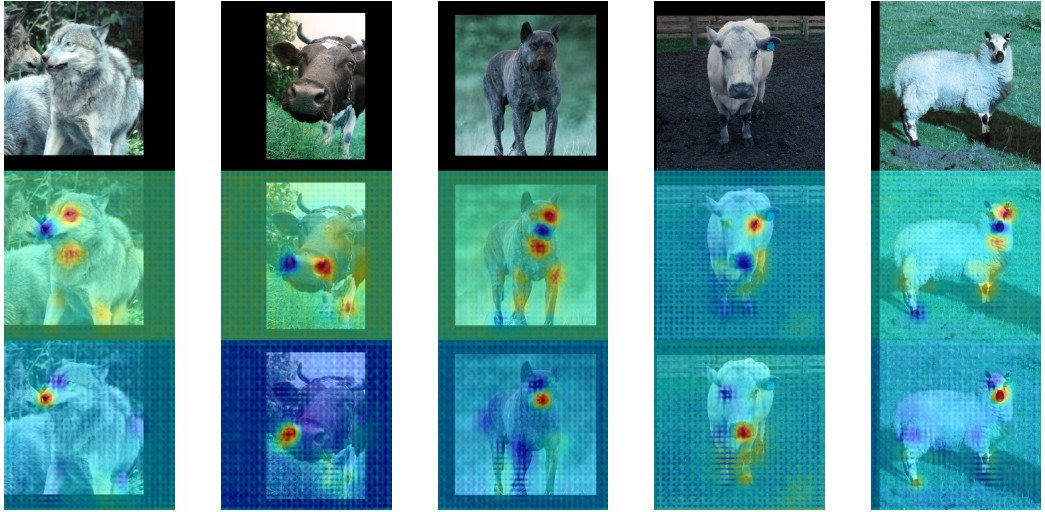

Figure 8: Attention weight visualization on AP-10K (Yu et al., 2021), where the bottom two rows correspond to the heatmap of different concepts. With a variety of animal species, *i.e.*, cow, wolf, sheep, and dog, the concept embedding can effectively attend and associate specific keypoints, which further justifies the effectiveness of the proposed approach.

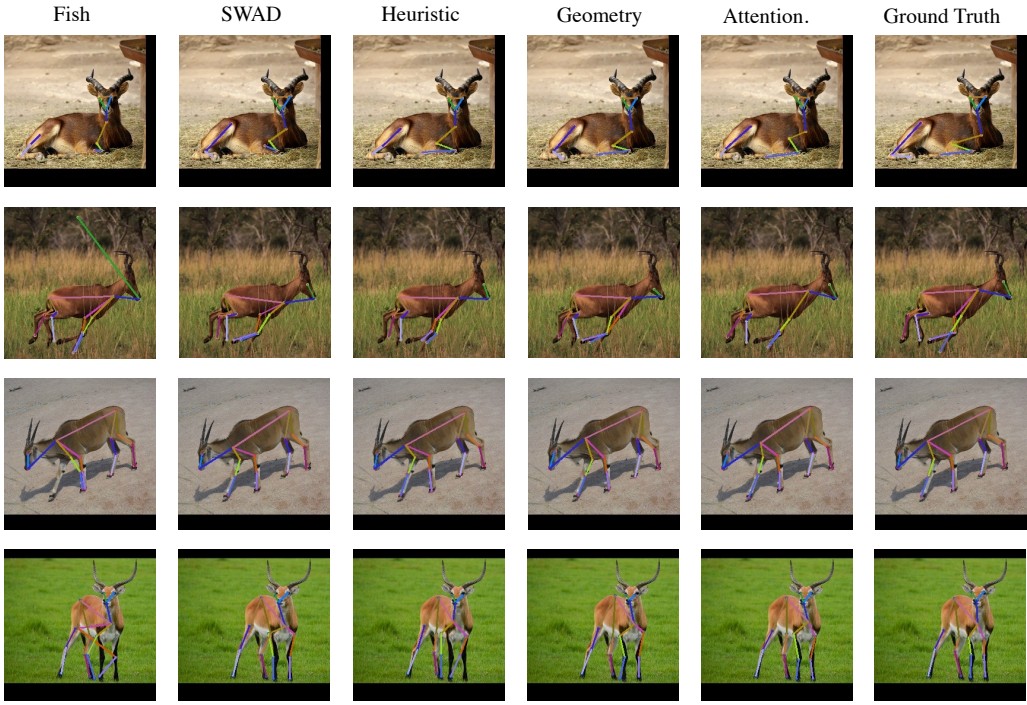

Figure 9: Pose estimation result under the intra-family DG on AP-10K (Yu et al., 2021). Experiments here follow the leave-one-out protocol on Family Bovidae and with the Antelope as the target domain. Compared with solutions, our method demonstrates stronger capability on joint localization and identification, especially on joints from non-rigid part, *e.g.*, legs.

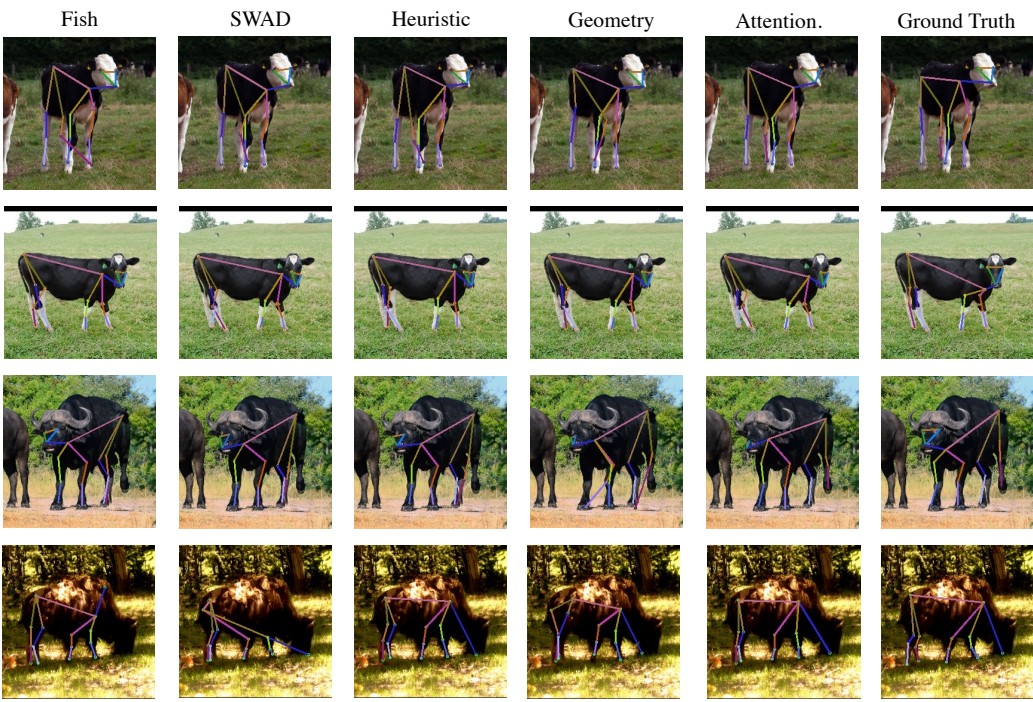

Figure 10: Pose estimation result under the inter-family DG on AP-10K (Yu et al., 2021). Experiments here follow the leave-one-out protocol and the target domain is Bovidae. Under a larger gap between species, our method maintains its superiority against previous solutions.

---

**Algorithm A** PyTorch-style pseudocode for pixel-to-concept attention

```python
import torch
import torch.nn as nn
import torch.nn.functional as F

class Pixel2Concept_Attention(nn.Module):
    def __init__(self, q_dim, k_dim, v_dim, hid_dim, out_dim, dropout=0.05):
        super(Pixel2Concept_Attention, self).__init__()
        # q_dim = k_dim = v_dim = hid_dim = out_dim = 256
        self.hid_dim = hid_dim
        self.w_query = nn.Linear(k_dim, hid_dim)
        self.w_key = nn.Conv2d(in_channels=q_dim, out_channels=hid_dim,
                      kernel_size=1, stride=1, padding=0)
        self.w_value = nn.Linear(k_dim, hid_dim)
        self.W       = nn.Conv2d(in_channels=hid_dim,out_channels=out_dim,
                      kernel_size=1, stride=1, padding=0)
        self.dropout = nn.Dropout(dropout)
        self.W = nn.Conv2d(out_dim, out_dim, kernel_size=1, padding=0)
        # Concept embedding, where k is the number of concepts
        self.conceptEmb =nn.Parameter(torch.FloatTensor(self.config.k, 256))

    def forward(self, feats):
        B, C, H, W = feats.shape
        q = self.w_query(self.concept_emb)
        k = self.w_key(feats)
        v = self.w_value(self.concept_emb)
        k = k.permute(0, 2, 3, 1)
        q = q.permute(1,0).unsqueeze(0)
        simi = torch.matmul(k, q)
        simi  = simi * (self.hid_dim**(-.5))
        simi = F.softmax(simi, dim=-1)
        simi = self.dropout(simi)
        out = torch.matmul(simi, v)
        out = out.permute(0, 3, 1, 2)
        out = feats + self.W(out)
        return out
```

---

## B  MORE IMPLEMENTATION DETAILS

**Experimental setup.** All the presented solutions are re-implemented with their released code.

`MixStyle` (Zhou et al., 2021): We find the default config works best: set the probability to 0.5, $\alpha$ is set to 0.3, and employ the random style augmentation.

`GradSur` (Mansilla et al., 2021): We adopt the default module, i.e., "PCGrad" function from the original code, which mitigates the gradient conflicts according to their inner product. No extra hyper-parameters is required to set.

`SWAD` (Cha et al., 2021): We report the best results from its presented variants, and no extra hyper-parameters are required to tune.

`Fish` (Shi et al., 2022): The meta-learning rate is set to 0.001. Other parameters for optimization are the same as our default setting.

**Training.** The training process consists of twos stages, i.e., the first stage derives the concept division with the learned concept embeddings, and the second stage optimizes with the concept-specific branches. To be specific, the first stage takes 60 epochs and the second takes 150 epochs. As for our one-stage solutions, we optimize them for 210 epochs. For the attention-based decomposition, the attention module is activated for both stages.

**Evaluation.** Following (Sun et al., 2019b; Cheng et al., 2020), we evaluate the pose similarities with OKS $= \frac{\sum_i \exp(-d_i^2/2s^2k_i^2)\delta(v_i>0)}{\sum_i \delta(v_i>0)}$. Here $d_i$ denotes the Euclidean distance between a detected keypoint and its corresponding ground truth, $v_i$ is the visibility flag of the ground truth, $s$ is the object scale, and $k_i$ is a per-keypoint constant that controls falloff. The reported average precision is the average of AP scores at OKS = (0.50, 0.55, ..., 0.90, 0.95.)

**Data augmentation.** In training, we employ the following data augmentation: random rotation ($[-30°, 30°]$), random scale ($[0.75, 1.5]$), random translation ($[-40, 40]$) to crop an input image patch of size $256 \times 256$ and random flip. In inference, only resize and normalization is performed.

**Attention module.** In Algo. A, we provide the implementation of the pixel-to-concept attention.

