# OpenReview forum: " Decompose to Generalize: Species-Generalized Animal Pose Estimation"
_ICLR.cc/2023/Conference — ICLR 2023 poster_

### Official Review · Reviewer_SQkn · 2022-10-23

**Confidence:** 4
**Correctness:** 4
**Technical Novelty And Significance:** 2
**Empirical Novelty And Significance:** 2
**Recommendation:** 5

**Clarity, Quality, Novelty And Reproducibility:**

The proposed method is well verified by the experiments.

Although the attention-based keypoint clustering method is interesting, the whole novelty is limited since the divide-and-conquer is a common approach In computer vision. Besides, specific characteristic of a species is not actually considered by the proposed work as claimed by the authors.

**Strength And Weaknesses:**

Strength:
1. The attention-based keypoint clustering method is interesting, and the effectiveness is verified well via comparison with SOTAs and ablation study.
2. Gradient conflict is used to analyze the effectiveness of the proposed idea.

Weaknesses:
1. The authors claim that some joint relations are consistent across all the species while some other joint relations are inconsistent and harmful. But the proposed method divides the keypoints into constant groups without considering species, which means the relations between keypoints are constant cross species. I think this work aims to build keypoint groups and estimate each group of keypoints relatively independently. This does not provide capability of handling  the diversity of species.
2. The main novelty lies in keypoint grouping, while the network-split's novelty is relatively limited. I expect the authors to perform multiple methods to encourage interaction between intra-group keypoints and suppress interaction between inter-group, in order to verify the effectiveness of the idea of keypoint grouping
3. The keypoint grouping result shown in Fig. 5 indicates that the grouping does not comply with prior knowledge very well. I doubt that this is relevant with training data's distribution.

**Summary Of The Paper:**

This paper proposes an animal-pose estimation method which can be applied to multiple animal species. Firstly the proposed method divides a set of animal keypoints into several groups. The keypoints in one group are supposed to have close relation, i.e. intra-group keypoints may provide a localization cue to each other. On the other hand, keypoints in one group may hinder the localization of the ones in another group due to the independence between groups. Then, the authors build a separate branch for each keypoint group for the keypoint detection network in order to encourage the interaction between intra-group keypoints and suppress the interaction between inter-group keypoints. The authors propose three grouping methods, among which two are based on rules and one is attention-based clustering method. The experimental results show that the proposed method using attention-based clustering outperforms some SOTAs , and thus the effectiveness the proposed method is verified.

**Summary Of The Review:**

The attention-based keypoint clustering method is interesting, but novelty is still not impressive. Besides, specific characteristic of a species is not actually considered by the proposed work as claimed by the authors.

---

> ### Author Response · Authors · 2022-11-19
> **Initial Response to Reviewer SQkn [part 1/2]**
>
> **A clarification before point-to-point response.**
>
> We would like to clarify two concepts that are easily confused with each other, i.e., consistent relation and close relation. We apologize for lacking highlight on their differences in the original manuscript, so that the confusion might cause you to raise two questions (Q1 and Q3).
>
> We recall that our key argument is: Consistent relation between joints is beneficial and should be encouraged. In the manuscript, we use the "eye and nose", which are close in all species as an intuitive example. However, we do not mean that consistent relation is a close relation. Instead, if two joints are far away for all the species, they are also considered as having consistent relations and are likely to be assigned into the same group (e.g., the Eye and the Hindleg in Fig. 6).
>
> We think this clarification can help you to better understand 1) why we do not use species-specific groups (Q1) and 2) why some grouping results seem to be counter-intuition (Q3). We provide detailed responses as below, and hope you will find our paper acceptable based on our responses.
>
> **Q1: The proposed method divides the keypoints into constant groups and thus results in constant relations across species. The specific characteristic of different species is not actually considered as claimed. Why not derive species-specific concepts?**
>
>
>
> We might not have explained our insight clearly enough so that you misunderstood our solution for the inconsistent relations. Using constant groups actually satisfies our motivation on both the consistent and inconsistent relations, simultaneously. Given that some relations are inconsistent across different species, our intention/objective is to discard the unsharable species-specific characteristics (while preserving the consistent relations). Therefore, we 1) assign the joints with consistent relations across species into the same group, and 2) separate the joints with inconsistent relations into different groups. These two operations jointly result in constant groups and actually comply with our motivation.
>
> In contrast, using specie-specific grouping (concepts) is contrary to the common sense[1][2] in domain generalization, i.e., the domain-specific pattern/knowledge should be discarded. This is because, in DG, the novel domain is not accessible for training. Therefore, using the domain-specific pattern as a prior for the novel domain actually compromises cross-domain performance. Our solution, i.e., separating joints with inconsistent relations into different groups, suppresses the domain-specific pattern and is thus beneficial.
>
>
> **Q2: More explorations/alternatives in the second stage except for the network split.**
>
> Thanks. Different from previous DG solutions that mainly focus on appearance or style variations, this paper studies a more challenging scenario that additionally includes higher dynamics of structure across different species. To this end, we investigate three solutions to discover the generalizable joint relationships drawn from various clues, i.e., visual, geometry, and anatomy, which have not been explored before. Then we devise a network split stage to strengthen the species-consistent relationships while mitigating the uncommon ones in an explicit manner. Especially, as you suggested, the network split can be replaced by other alternatives for modulating the joint relationships, hence offering extra compositionality. And we will leave it in the future work for exploration.
>
> **Q3: Some results of the keypoint grouping in Fig. 6 (Fig. 5 of the original paper) do not comply with prior knowledge very well. Is this due to the distribution of training data?**
>
> The geometry-based and attention-based grouping results indeed have significant differences and even some conflict against human intuition. However, this phenomenon is reasonable and should NOT be considered a weakness. While human intuition is prone to semantic partition (as shown in the "heuristic" strategy in Fig. 6), the geometry-based and attention-based strategies are more data-driven (as you mentioned). Experimental results (Table 1 and 2 in the manuscript) show that the geometry-based / attention-based strategy achieves comparable / higher results, compared with the heuristic one. We thus infer that the conflict against human intuition is reasonable.
>
> Moreover, we have an in-depth reason why the counter-intuition happens. As we clarified at the beginning, in our method, a single group should contain joints with "consistent relation" rather than "close relation". Therefore, if two joints are far away (either spatially or semantically far) across all the species, they are likely to be assigned to the same group. Assigning consistently-far joints (e.g., the eye and hindleg) into the same group seems to be counter-intuitive at the first glance but is actually reasonable, according to conceptual analysis and the empirical results.

---

> ### Author Response · Authors · 2022-11-19
> **Initial Response to Reviewer SQkn [part 2/2]**
>
> **Q4. The novelty is limited since the principle of ''divide and conquer'' is common in computer vision.**
>
> We respectively disagree with this point. Though the "divide and conquer" paradigm is popular, our method is still novel. Our key novelty and contribution is that we find the joint relation is critical to cross-species pose estimation, i.e., 1) the consistent relation is beneficial while 2) the inconsistent relation hampers cross-species generalization. Based on these two insights, the "divide and conquer" is only the simple tool (yet effective) to deliver our strategy.
>
>
>
>
> [1] Ben-David et.al. A theory of learning from different domains. Machine learning, 2010.
>
> [2] Zhou et.al. Domain generalization: a survey. IEEE TPAMI, 2022.

---

### Official Review · Reviewer_e7pK · 2022-10-24

**Confidence:** 4
**Correctness:** 3
**Technical Novelty And Significance:** 2
**Empirical Novelty And Significance:** 2
**Recommendation:** 6

**Clarity, Quality, Novelty And Reproducibility:**

The motivation in this article is well-explained and generally reasonable, but the technical novelty in the individual parts are limited.

**Strength And Weaknesses:**

Pros:

-The motivation in this paper is reasonable in that some of the relationships between specific joints may be shared by multiple animal species; hence, considering such relationships aids in cross-species animal pose estimation.

Cons:

-K (Number of concept embedding) is a fixed number and is set to 3 for all tasks. Concept embedding in Fig 3 (b) is similar to the codebook[1]. When estimating the poses of various animal species, it is better to select a larger value for k and let the network dynamically select the concept embeddings to utilize.

-The gap between the source and target domain is relatively small in both intra-family and inter-family (basically a four-legged mammal). A larger domain gap would be more convincing (e.g., fish, birds).

-Lack experiments on existing large-scale animal pose datasets, e.g. [2]

【1】 Van Den Oord A, Vinyals O. Neural discrete representation learning[J]. Advances in neural information processing systems, 2017, 30.

【2】 Ng, Xun Long, et al. "Animal Kingdom: A Large and Diverse Dataset for Animal Behavior Understanding." Proceedings of the IEEE/CVF Conference on Computer Vision and Pattern Recognition. 2022.

**Summary Of The Paper:**

This paper proposes a domain generation approach, D-Gen, for cross-species animal pose estimation. D-Gen first learns k concept embeddings using an attention-based module. Then, previously-learned concept embeddings are used to decompose the body joints into k joint concepts with nearest neighbor search. Finally, D-Gen split the top layer of the network into multiple concept-specific branches based on the joint concepts. Experiments are conducted in both intra-family and inter-family settings to show the effectiveness of D-Gen.

**Summary Of The Review:**

The paper addresses an interesting problem but is hampered by the limited novelty of the approach and the lack of experiments for cross-species animal pose estimation with a larger domain gap(e.g. fish, birds).

---

> ### Author Response · Authors · 2022-11-19
> **Initial Response to Reviewer e7pK**
>
> **Q1: The number of concepts $k$ is set to 3 for all tasks. Since concept embedding is similar to the codebook [1], using a larger $k$ to allow higher dynamics for learning joint relations might be better.**
>
> We find the manuscript lacks a detail so that you over-estimated the optimal value for $k$: there are only 17 key points in total (this detail appeared only once in Section 4.1 Setup in the manuscript). We apologize for our careless arrangement and will highlight this detail in the method part, as well.
>
> Moreover, our choice of setting $k=3$ is based on experimental results (Section 4.3 and Fig. 5 (a) (b) in the manuscript): further increasing $k$ decreases the accuracy for both intra-family and inter-family scenarios.
>
>
> **Q2: Difference with VQ-VAE [1].**
>
>
> The concept embeddings in this paper are different from the codebook of VQ-VAE in terms of both motivation and technique. First, VQ-VAE seeks to derive discrete representations to replace the continuous features, thus offering extra flexibility for representation learning. In contrast, in our method, the concept embeddings are designed to bridge and discover inter-joint relationships. Second, technically, VQ-VAE adopts a reparameterized process to detach the gradient backpropagation between the model and the codebook while our method does not.
>
> **Q3: Experiments on Animal Kingdom [2].**
>
>
> Good suggestion. Compared with AP-10K and Animal Pose Dataset, the animal kingdom dataset can provide a larger domain gap and is thus more challenging. During rebuttal, we add experiments on this dataset and have already obtained the results for the two most recent state-of-the-art methods (SWAD and Fish), as well as ours. Although these results are still incomplete due to limited resources and time, the already-achieved results provide strong evidence for the superiority of our method. For example, compared with the strongest competing method Fish [3], our method achieves +1.7 % higher PCK score. These experimental results are added to Table 4 in the revised manuscript and will be further full-filled upon acceptance.
>
> [1] Van Den Oord A, Vinyals O. Neural discrete representation learning. Advances in neural information processing systems, 2017, 30.
>
> [2] Ng, Xun Long, et al. "Animal Kingdom: A Large and Diverse Dataset for Animal Behavior Understanding." Proceedings of the IEEE/CVF Conference on Computer Vision and Pattern Recognition. 2022.
>
> [3] Shi et.al. Gradient matching for domain generalization. ICLR 2022.

---

> > ### Comment · Reviewer_e7pK · 2022-12-03
> > **Acknowleding author responses**
> >
> > Dear Authors,
> >
> > I have noticed that you have added experiments on the Animal Kingdom to Table 4, and the results obtained in the intra-family setting show that the method is equally superior for larger domain gaps.
> > Although I still think it's not good that there will only be one concept label for each keypoint, which severely limits the idea of keypoint grouping,I agree with reviewer 2nfE that the method is novel in this particular area.
> > I have therefore decided to raise the score to 6.

---

> > > ### Author Response · Authors · 2022-12-04
> > > **Thank you**
> > >
> > > Dear Reviewer e7pK,
> > >
> > > Thanks for your positive assessment and constructive feedback on our work. We gratefully acknowledge that your suggestions helped us to improve our manuscript.
> > >
> > >
> > > Best,
> > >
> > > Authors

---

### Official Review · Reviewer_2nfE · 2022-10-24

**Confidence:** 3
**Correctness:** 3
**Technical Novelty And Significance:** 4
**Empirical Novelty And Significance:** 4
**Recommendation:** 8

**Clarity, Quality, Novelty And Reproducibility:**

The manuscript is clear and the experiments are well described. The network design appears novel.

**Strength And Weaknesses:**

Strengths
1) Attention Approach is novel, and compared against several other reasonable baselines and datasets. The experiments are well documented and well performed. I think the advantages are somewhat appreciable in the intra-family case, and less so in the inter-family case, but this will be a nice result to build on.
2) Manuscript is clear and easy to follow, trained on standard benchmarks in the field and fairly comprehensive. I appreciated the didactic figure 5
3) I thought the gradient conflict analysis was interesting and a nice empirical motivation for the work.


Weaknesses
1) Treatment of uncertainty in tables. I am not sure how large the uncertainty is for the experiments in Tables 1,2,3 are. It would help to have reporting of this to evaluate the significance of some of the results seen – I am not sure how seriously to take 0.5 differences in AP. The error bars reported below each family also seem too low given the spread of values (e.g. Table 1, attention/average has a s.e.m of 0.5 ?). These could be computed from different model seeds or training data subsets.

2) The oracle shows worse performance than the domain generalization approaches at times in Table 1. Is this because of a lack of data for rarer classes (e.g. king cheetah)? In these cases the oracle would have much less training data. It might make sense to also try training the oracle on all datasets in the family, and report the number of instances of each animal.
3) Its worth noting that none of the approaches seem very good on the inter-family experiments (Tables 2 and 3)
4) It would help potential users to have representation qualitative visualization of some of these performance differences. What does a mAP gain of 63 vs 59 look like for Bobcat in Table 1? What are typical inter-family results? Without these it is hard for prospective users to gauge the quality of the performance gains.
5) Why simply average the loss with the baseline network and why not include a learnable weight for combining the predictions?

Comments:
 I think the approach could have more broad generality, e.g. for facial keypoint tracking where separate groups of keypoints co-vary.
I would be curious if any of the domain adaptation approaches can be combined, e.g. those that seek to match distributions should be fairly orthogonal to the present work.


Nits
Figure 1:  What are the methods for the bar graphs on the left, why are there no error bars?
Table 1: A.S. undefined.
Viewpoint dependence. Could conceive of as a style difference.
Figure 6 and 5 switched
Its worth noting that these 2D style transfer approaches will be sensitive to camera viewing angle, and the relation between joints and joint concepts may depend on viewing angle.
Why is the domain generalization performance so poor for panda bears and polar bears?
Can you provide more detail on the empirical risk minimization approach.


**Summary Of The Paper:**

This paper describes a new approach for domain generalization in cross-species pose estimation using different approaches to partition joints to be estimated into different ‘concepts’, and separately training parts of the network to estimate the position of each joint concept group. Three approaches for separating joint concepts are used: based on the proximity of joints to one another (heuristic), based on the joint affinity matrix (geometric), and based on a learned attention mechanism (attention). In each images are processed using common feature extraction layers, and then joint concepts are separated and processed in separate branches. The manuscript introduces the details of the attention based approach, which is novel and shown to perform superior to the others in several cases. They show that the attention approach minimizes gradient conflicts, i.e. does a better job of separating weight gradient vectors across joint groups.
They then perform benchmark experiments on two datasets that allow for testing domain generalization approaches within and across animal family. They compare the performance of each approach, along with a random split to account for ensemble gains, to competing domain generalization approaches and oracle approaches, in particular training and testing within a species or family. In many cases the attention approach performs superior to other domain generalization approaches. They close by investigating the effect of salient hyperparameters: the number of layers that are concept-specific and the number of concepts.


**Summary Of The Review:**

Overall I thought this was an interesting approach and seemed to show some promise compared to other approaches. The manuscript was clear and contained both fairly thorough benchmark comparisons, some hyperparameter optimization and some explanations for performance gains. I have reservations about the overall effect size/improvement and variance that I would like to see addressed, but otherwise I think it is a good contribution.

---

> ### Author Response · Authors · 2022-11-19
> **Initial Response to Reviewer 2nfE  [part 1/2]**
>
> **Q1: The error bars seem too low. Please clarify the treatment of uncertainty.**
>
>   We already use different random seeds and repeat each experiment three times, following the setup in [1]. Our error bars (uncertainty results) are close to the error bars in [1], as well. Therefore, we think there should be no concerns about whether the error bars are abnormal.
>
>   That being said, we take your suggestion on using different training data subsets seriously. During rebuttal, we adopt two more new data splits. Consequently, we repeat our experiments on two new data splits with three different random seeds. Under such variation, we re-evaluate the uncertainty in two scenarios, i.e., intra-family DG within Bovidae and inter-family DG on AP-10K. The results are summarized and added to the appendix (Table 12), and show consistent uncertainty with the main paper on both benchmarks.
>
>
> **Q2: Why does the Oracle sometimes show worse performance in Table 1? Is it because the Oracle has less training data? Another Oracle uses all the source + target data.**
>
> Thanks for this question. The poor oracle performance is indeed due to less training data. Concretely, the Oracle is trained on the target domain (with only one specie) and sometimes has very limited training samples, e.g., the family king cheetah only has 22 training samples. In contrast, the evaluated methods are trained on the source domain (consisting of multiple species) and usually have more training data.
>
> According to your suggestion, we add another Oracle (Oracle-Full) to Table 1 and 2 of the revised manuscript. Oracle-Full combines both the source and target domain for training. Its accuracy is relatively high and may be viewed as the upper bound of cross-species pose estimation. Oracle-Full on other datasets will be updated in the final version.
>
> **Q3: It is worth noting that none of the approaches seem very good in the inter-family experiments (Tables 2 and 3).**
>
> Thanks. This phenomenon is reasonable because inter-family DG is inherently much more difficult than intra-family DG. Specifically, the domain gap between different families is larger than the domain gap between different species. Given the larger domain gap, cross-domain generalization is generally more difficult.
>
> **Q4: Qualitative visualization for a better understanding of the cross-domain pose estimation results and the performance gains.**
>
> Thanks for this good suggestion. During rebuttal, we visualize some cross-domain pose estimation examples for both the baseline and our method. We have added these visualizations to the appendix (Fig. 9 and Fig. 10).
>
> **Q5: Why simply average the losses? How about employing learnable weights for fusing losses in Eq.4?**
>
> Thanks for these two questions.
>
> 1) We simply average the losses (for training the baseline and the concept-specific branches) because we found that our method is largely robust to the loss weights. Specifically, in our preliminary experiments, we have investigated the impact of varying the loss weights from [0.1:1] to [5.0:1] on intra-family DG on family Bovidae from AP-10K. Within this range, the variation of the achieved mAP is only around 0.7. Therefore, we adopt the 1:1 loss weights for simplicity.
>
> 2) That being said, we agree that using learnable loss weights is likely to bring further improvement. However, we presently do not have a good idea of how to learn the weights. Given that using the 1:1 loss weights achieves acceptable results and already demonstrates our key argument (i.e., decomposing the joints benefits cross-domain generalization), we sincerely hope you will allow us to leave the learnable weights problem to our future work.
>
> **Q6: Can the proposed method be combined with other DA/DG methods, e.g., those seek to match distributions?**
>
> Thanks. We think combining our method with other DA/DG methods has the potential of obtaining complementary benefits. For example, if there exists a domain shift regarding the lightning condition, our method might not be able to improve this specific factor, because it lays emphasis on the joint relationships. Under this condition, additionally using an AdaIn-based DG method (e.g., DSU [2]) is likely to bring extra benefits by suppressing the lightning condition shift.

---

> ### Author Response · Authors · 2022-11-19
> **Initial Response to Reviewer 2nfE [part 2/2]**
>
> **Q7: Some editorial problems.**
>
> Thanks. We have revised all the problems that you mentioned and will carefully scrutinize the whole manuscript.
>
> **Q8: Discussion on the gap induced by the difference in viewpoint or styles.**
>
> Insightful suggestion. The mentioned two issues are indeed two critical challenges for modeling inter-joint relationships. For example, the viewpoint shift can undermine the geometry-based decomposition as it relies on the pure geometric distance in the 2D images. In contrast, the heuristic decomposition draws the empirical knowledge of geometry but totally neglects the relationships upon appearances or styles. Compared with the two, the attention-based solution can overcome them to some extent as it considers the inter-joint relationships from two aspects, i.e., visual and geometry, thus being less susceptible to these two issues.
>
> **Q9: Why is the DG performance so poor for panda bears and polar bears?**
>
> We infer it is because when the target domain is the panda bear / polar bear, the domain gap is inherently large. Concretely, in the Ursidae family, the panda is the only species in mixed color (black and white), while other bears are roughly in a single color. Therefore, generalizing the model from the single-color bears to pandas is particularly difficult. Similarly, the DG from dark-color bears (black bear and brown bear) to pure-white color bears (polar bear) is difficult.
>
>
> [1] Yu et.al AP-10K: A Benchmark for Animal Pose Estimation in the Wild. NeurIPS 2021 Datasets and Benchmarks Track.
>
> [2] Li et.al. Uncertainty Modeling for Out-of-Distribution Generalization. ICLR 2022.

---

### Official Review · Reviewer_uMhs · 2022-10-25

**Confidence:** 4
**Correctness:** 4
**Technical Novelty And Significance:** 3
**Empirical Novelty And Significance:** 3
**Recommendation:** 6

**Clarity, Quality, Novelty And Reproducibility:**

The paper is well written and easy to read. The description of the method is sufficiently clear. Nevertheless, some aspects like inference (see above) could be described in more detail.

Although the approach is based on established solutions (domain generalization, network splitting, etc.) their application to pose estimation, and animal pose estimation in particular, is novel.

Regarding reproducibility, a code snippet is provided that can help in implementing the method and reproducing the results.

**Strength And Weaknesses:**

## Strengths
The proposed approach is principled and well motivated. Moreover, there are improvements in the animal pose estimation performance, regardless of the decomposition method, i.e. attention-based, heuristic or geometric. It is also important that the attention-based performs significantly better with respect to the other two methods. The network split is also well motivated based on gradient analysis.
The method is compared against state-of-the-art domain generalization methods.

## Weaknesses
The fact that the proposed approach is based on two distinct stages, namely joint decomposition and network split, can be considered a weakness. This is mainly because the two steps appear to be strictly coupled, yet they are treated independently. It would not be a weakness if being disjoint the two stages could offer better compositionality (e.g. some step could be replaced with some alternative) but this is not discussed in the paper.

Moreover, the use of the split network seems to require some relatively involved bookkeeping for composing the per-concept features. A short discussion on this subject could help reproducibility.

Additionally, inference is only briefly discussed in the paper, mainly in Figure 2. It would help if more details were provided.

**Summary Of The Paper:**

This paper proposes a method for animal pose estimation able to generalize to different species. The method approaches the problem from the domain generalization point of view. It is based on the observation that some relations among joints remain consistent among different species, while others change drastically. According to this, the paper proposes to separate joints to concepts/groups and process them using a network with split top layers, as to break inconsistent relations while maintaining consistent ones.

**Summary Of The Review:**

Based on the discussion above, I think that the proposed method proposes an interesting and significant contribution to the less explored problem of animal pose estimation (in comparison to human pose estimation). I propose acceptance, although there are some aspects which could be improved (see weaknesses above)

## Comments after the rebuttal
I find the proposed approach novel in the specific domain. The contributions are well motivated and interesting, especially the distinction between consistent and not-consistent joints is intuitive and supported by the results. Nevertheless, the type of consistency needs to be better defined to avoid confusion (e.g., geometric/topological vs semantic consistency).

I am quite satisfied with the author responses and I think that the paper has become stronger after the revisions. However, the increase in performance between different animal families is somewhat limited. I keep my score to 6 as I think that the paper is acceptable for publication.

---

> ### Author Response · Authors · 2022-11-19
> **Initial Response to Reviewer uMHs**
>
> **Q1: Separating the joint decomposition and the network split into two independent stages seems not very elegant. It would not be a weakness if being disjoint these two stages could offer better compositionality.**
>
>
>
>
> Thanks for this insightful suggestion. There are two main reasons:
> 1) Separating these two stages brings better compositionality (as you mentioned), so that we may easily replace each step with some alternative.
> 2) Separating these two stages allows us to wait until the first stage (joint decomposition) fully converges, so that the decomposition results are more stable and reliable. In contrast, if we compile these two steps into a single end-to-end stage, we have to use the intermediate decomposition results, which are not good enough.
>
>
> **Q2: Clarification on 1) the process of composing per-concept features and 2) the inference pipeline in Fig.2 (right).**
>
>
> 1) The details of inference are provided in Sec 3.3 (below Eq. 4). We have highlighted it and complemented it with more details regarding composing the joint features from the concept-specific branches.
> 2) Specifically,  we first concatenate the output of concept-specific branches and average them with the baseline branch to obtain the final output. Notably, before the concatenation, we first sort them according to their original joint index as the joints are not sequentially divided. The full code will be publicly released upon acceptance.

---

### Author Response · Authors · 2022-11-19
**A Summary of Revisions**

We sincerely thank all reviewers for their time and effort in reviewing this paper.

We have revised our manuscript based on your comments. We use blue color to highlight these revisions in the manuscript and summarize the main changes as below:


* Section 1: We add error bars for Fig. 1.
* Section 3.3: We explain the inference pipeline in more detail.
* Section 4.2: We explain the details of the Oracle and ERM baseline, and introduce a stronger oracle method (Oracal-Full) trained with "source + target". We add one more dataset, i.e., Animal Kingdom for evaluation (Table 4).
* Section 4.3: We switch the order of Fig.5 and Fig.6.
* Appendix. A: We report results with random runs on new data split to better evaluate the uncertainty (Table 12).  We add qualitative results for a better understanding of the improvement (Fig.  9 and Fig. 10).

---

### Decision · Program_Chairs · 2023-01-20

**Decision:**

Accept: poster

**Justification For Why Not Higher Score:**

Some ablation studies are not included, e.g., performing multiple methods to encourage interactions between intra-group key-points and suppress interactions between inter-group.

**Justification For Why Not Lower Score:**

The proposed method is novel, and the experiments have supported the efficacy.

**Metareview: Summary, Strengths And Weaknesses:**

This paper addresses the problem of cross-animal pose estimation (generalization to novel animals). The authors propose a Decompose-to-Generalize method to break the inconsistent relations (relations that are inconsistent for different species due to species variation and bring distraction) while preserving the consistent ones (relations that are consistent across all species). Reviewers think the method is novel and interesting. Reviewer uMhs: Approach is principled and well motivated. Network split is also well motivated based on gradient analysis. Reviewer 2nfE: Gradient conflict analysis was interesting. Attention Approach is novel. Though Reviewer e7pK commented that paper has limited novelty, but during the AC-Reviewer discussion,  Reviewer e7pK pointed out that keypoint grouping is novel in this field, and the effectiveness of keypoint grouping has been supported in experiments. Reviewer SQKn gives negative score, but also mentioned that the attention-based keypoint clustering method is interesting. Overall, the paper is clearly presented, and the efficacy is supported by the experiments. The weaknesses include the relatively marginal performance improvement and the lack of some ablation studies as mentioned by reviewer SQKn. Considering the novelty of the work, the AC recommends acceptance. Authors need to improve the final version by incorporating the promised revisions to the paper, and add the ablation studies (e.g., performing multiple methods to encourage interaction between intra-group key-points and suppress interaction between inter-group).

**Note From Pc:**

if the above contains the word "oral" or "spotlight" please see: "oral" presentation means -> notable-top-5% and "spotlight" means -> notable-top-25%. As stated in our emails, we are disassociating presentation type from AC recommendations

**Summary Of Ac-Reviewer Meeting:**

During discussion, Reviewers uMhs, 2nfE, and e7pK all think the paper is novel and interesting, and recommend accept. Though Reviewer  SQKn does not reply for the discussion, his/her main concerns have also been well addressed by authors to a large extent. Reviewer SQkn also pointed out that in the proposed network split, a keypoint can only belong to one group, which would limit the possible combinations of joints. This may be one drawback of this paper. But keypoint grouping is still novel in the field of animal pose domain generation, and the effectiveness of keypoint grouping has been well supported in experiments, which shows the proposed domain generation method is superior to other methods in the field. Reviewer 2nfE also highlighted that the novelty of the approach is important and all reviewers agree this paper is novel. The AC agrees this and recommends accept.